

# Tree phyllosphere bacterial communities: exploring the magnitude of intra- and inter-individual variation among host species

Isabelle Laforest-Lapointe[1,2], Christian Messier[1,2,3] and Steven W. Kembel[1,2]

[1] Centre d'étude de la forêt, Montreal, Canada
[2] Sciences Biologiques, Université du Québec à Montréal, Montreal, Quebec, Canada
[3] ISFORT - Institut des Sciences de la Forêt Tempérée, Université du Québec en Outaouais, Ripon, Quebec, Canada

Corresponding author
Isabelle Laforest-Lapointe,
isabelle.laforest.lapointe@gmail.com

## ABSTRACT

**Background**. The diversity and composition of the microbial community of tree leaves (the phyllosphere) varies among trees and host species and along spatial, temporal, and environmental gradients. Phyllosphere community variation within the canopy of an individual tree exists but the importance of this variation relative to among-tree and among-species variation is poorly understood. Sampling techniques employed for phyllosphere studies include picking leaves from one canopy location to mixing randomly selected leaves from throughout the canopy. In this context, our goal was to characterize the relative importance of intra-individual variation in phyllosphere communities across multiple species, and compare this variation to inter-individual and interspecific variation of phyllosphere epiphytic bacterial communities in a natural temperate forest in Quebec, Canada.

**Methods**. We targeted five dominant temperate forest tree species including angiosperms and gymnosperms: *Acer saccharum*, *Acer rubrum*, *Betula papyrifera*, *Abies balsamea* and *Picea glauca*. For one randomly selected tree of each species, we sampled microbial communities at six distinct canopy locations: bottom-canopy (1–2 m height), the four cardinal points of mid-canopy (2–4 m height), and the top-canopy (4–6 m height). We also collected bottom-canopy leaves from five additional trees from each species.

**Results**. Based on an analysis of bacterial community structure measured via Illumina sequencing of the bacterial 16S gene, we demonstrate that 65% of the intra-individual variation in leaf bacterial community structure could be attributed to the effect of inter-individual and inter-specific differences while the effect of canopy location was not significant. In comparison, host species identity explains 47% of inter-individual and inter-specific variation in leaf bacterial community structure followed by individual identity (32%) and canopy location (6%).

**Discussion**. Our results suggest that individual samples from consistent positions within the tree canopy from multiple individuals per species can be used to accurately quantify variation in phyllosphere bacterial community structure. However, the considerable amount of intra-individual variation within a tree canopy ask for a better understanding of how changes in leaf characteristics and local abiotic conditions drive spatial variation in the phyllosphere microbiome.

## INTRODUCTION

The phyllosphere microbiota represents the communities of microorganisms including bacteria, archaea, and eukaryotes such as fungi that are associated with plant leaves (*Inácio et al., 2002*; *Lindow & Brandl, 2003*). Phyllosphere microbes influence host fitness through a variety of mechanisms such as plant hormone production and protection from pathogen colonization (*Innerebner, Knief & Vorholt, 2011*; *Ritpitakphong et al., 2016*). As a result of their effect on host plant fitness, leaf microorganisms can influence plant population dynamics and community diversity (*Clay & Holah, 1999*; *Bradley, Gilbert & Martiny, 2008*) as well as ecosystem functions including water (*Rodriguez et al., 2009*) and nutrient cycling (*Van Der Heijden, Bardgett & Van Straalen, 2008*; *McGuire & Treseder, 2010*; *Allison & Treseder, 2011*). Tree microbial phyllosphere communities have been studied in tropical (*Lambais et al., 2006*; *Lambais, Lucheta & Crowley, 2014*; *Kim et al., 2012*; *Kembel et al., 2014*; *Kembel & Mueller, 2014*), temperate (*Jumpponen & Jones, 2009*; *Redford & Fierer, 2009*; *Redford et al., 2010*; *Jackson & Denney, 2011*) and Mediterranean forests (*Penuelas et al., 2012*), along altitudinal gradients (*Cordier et al., 2012a*; *Cordier et al., 2012b*), and in deserts (*Finkel et al., 2011*; *Finkel et al., 2012*). In order to understand the structure and function of phyllosphere microbial communities, studies typically either assume that a single sample of leaves from a plant canopy is representative of the phyllosphere community of the entire tree or host species (*Lambais et al., 2006*; *Kim et al., 2012*; *Kembel et al., 2014*), or control for spatial structure in phyllosphere community structure by mixing leaves from multiple canopy locations (*Redford & Fierer, 2009*; *Redford et al., 2010*; *Jumpponen & Jones, 2009*; *Jumpponen & Jones, 2010*; *Finkel et al., 2011*; *Finkel et al., 2012*; *Cordier et al., 2012a*; *Cordier et al., 2012b*). In this study our aim was to quantify the relative importance of intra-individual versus inter-individual and inter-specific variation in the structure of temperate tree phyllosphere communities across multiple host species.

Host genetic factors (*Bodenhausen et al., 2014*; *Horton et al., 2014*) and taxonomic identity (*Redford et al., 2010*; *Kembel et al., 2014*) are important drivers of phyllosphere bacterial community structure. Most studies of phyllosphere communities across different host species have assumed within-plant and within-species variation in phyllosphere community structure to be negligible, and looked passed intra-individual and inter-individual variation (but see *Redford et al. (2010)* and *Leff et al. (2015)*). In tree phyllosphere studies, samples are usually taken from shade leaves either at the bottom of the canopy or at mid-canopy height near the trunk. However, the technique to sample phyllosphere communities vary between studies, ranging from studies that sampled leaves from a specific canopy location (i.e., *Kembel et al., 2014*; *Kembel & Mueller, 2014*) to taking multiple leaves from around the canopy at the same height (i.e., *Redford & Fierer, 2009*; *Redford et al., 2010*; *Jackson & Denney, 2011*). However, *Leff et al. (2015)* demonstrated for a single tree species

(*Ginkgo biloba*) that there is intra-individual variation in phyllosphere community structure within the canopy of a single tree. The relative importance of this within-individual variation versus inter-individual and inter-specific variation, and the degree to which a sample of leaves from a canopy are representative of the microbiome of an individual or a species, is not well understood.

A multitude of factors could influence microbial community structure on leaves within a tree canopy. Leaf position in the canopy defines the degree of exposure to ultraviolet radiation and wind and therefore community structure could change depending on the position of the leaves sampled. Exposure to ultraviolet radiation has been shown to increase the diversity of the maize leaf microbial community (*Kadivar & Stapleton, 2003*) and anoxygenic phototropic bacteria have been detected in the phyllosphere of *Tamarix nilotica* (*Atamna-Ismaeel et al., 2012*). This phenomenon could also be caused by leaf morphological and ecophysiological attributes associated with high light availability (thicker leaves, lower specific leaf area, lower water content, higher total chlorophyll, higher photosynthetic activity rate; *Lichtenthaler et al., 1981*). Variation in atmosphere conditions within the canopy (i.e., increased exposure to wind and gas exchange levels) modifies local leaf humidity conditions potentially influencing leaf epiphytic bacterial communities by inhibiting or favoring the growth of particular groups (*Medina-Martínez et al., 2015*). Wind exposure could reduce leaf moisture and induce a stomata closure (*Grace, Malcolm & Bradbury, 1975*), which could impact the diffusion of nutrients and reduce the size of microbial aggregates (*Leveau & Lindow, 2001*; *Miller et al., 2001*).

In this study, we aim to (1) compare the intra-individual, inter-individual and interspecific variation of phyllosphere bacterial communities; (2) characterize the composition of epiphytic phyllosphere bacterial communities at different canopy locations for five tree species; and (3) make practical recommendations for the sampling of tree phyllosphere bacterial communities. We hypothesized that (1) the magnitude of intra-individual variation will be smaller than inter-individual and interspecific variation, (2) that canopy location will be a significant driver of phyllosphere bacterial community structure because of variation in abiotic conditions (e.g., radiation, wind), and changes in ecophysiological and morphological leaf characteristics.

## MATERIALS AND METHODS

### Study site & host-tree species

The two study sites are located in a natural temperate forest stand in Gatineau (45°44′50″N; 75°17′57″W) and Sutton (45°6′46″N; 72°32′28″W) Quebec, Canada. These sites are characterized by a cold and humid continental climate with temperate summer. A total of six individuals (three at each site) from each of five tree species common to temperate forests and dominant in the canopy were sampled to provide representatives of both angiosperms and gymnosperms: *Abies balsamea* (Balsam fir), *Acer rubrum* (Red maple), *Acer saccharum* (Sugar maple), *Betula papyrifera* (Paper birch) and *Picea glauca* (White spruce).

## Bacterial community collection

We sampled phyllosphere communities from trees on August 29, 2013 as part of another experiment (*Laforest-Lapointe, Messier & Kembel, 2016*). Sampling was carried out one week after the last rainfall event. We defined three strata within the canopy: bottom-canopy (1–2 m height), mid-canopy (2–4 m height), and top-canopy (4–6 m height). 30 individuals were randomly selected by picking random geographic coordinates and finding the closest individual at this location. For the first tree sampled from each species, we clipped 50–100 g of leaves at the four cardinal points at mid-canopy height, plus a single sample at bottom-canopy and top-canopy heights, into sterile roll bags with surface-sterilized shears. We also sampled bottom-canopy leaves from two other randomly chosen trees from each species. For bacterial community collection and amplification we used the protocols described by *Kembel et al. (2014)*. We collected microbial communities from the leaf surface by five minutes of horizontal mechanical agitation of the samples in a diluted Redford buffer solution. We resuspended cells in 500 µL of PowerSoil bead solution (MoBio, Carlsbad, California). We extracted DNA from isolated cells using the PowerSoil kit according to the manufacturer's instructions and stored at −80 °C.

## DNA library preparation and sequencing

We used a two-step PCR approach to prepare amplicon libraries for the high-throughput Illumina sequencing platform. The use of combinatorial primers for paired-end Illumina sequencing of amplicons reduced the number of primers while maintaining the diversity of unique identifiers (*Gloor et al., 2010*). First, we amplified the V5–V6 region of the bacterial 16S rRNA gene using chloroplast-excluding primers in order to eliminate contamination by host plant DNA (16S primers 799F-1115R (*Redford et al., 2010*; *Chelius & Triplett, 2001*)) following protocols described by *Kembel et al. (2014)*. We cleaned the resulting product using a MoBio UltraClean PCR cleanup kit. We isolated a ∼445-bp fragment by electrophoresis in a 2% agarose gel, and recovered DNA with the MoBio GelSpin kit. We prepared multiplexed 16S libraries by mixing equimolar concentrations of DNA, and sequenced the DNA library using Illumina MiSeq 250-bp paired-end sequencing at Genome Quebec.

We processed the raw sequence data with PEAR (*Zhang et al., 2014*) and QIIME (*Caporaso et al., 2010*) software to merge paired-end sequences to a single sequence of length of 350 bp, eliminate low quality sequences (mean quality score < 30 or with any series of 5 bases with a quality score < 30), and de-multiplex sequences into samples. We eliminated chimeric sequences using the Uclust and Usearch algorithms (*Edgar, 2010*). Then, we binned the remaining sequences into operational taxonomic units (OTUs) at a 97% sequence similarity cutoff using the Uclust algorithm (*Edgar, 2010*) and determined the taxonomic identity of each OTU using the BLAST algorithm (Greengenes reference set) as implemented in QIIME (*Caporaso et al., 2010*). The number of sequences per sample ranged from 6,256 to 75,412. From these 1,499,777 sequences, we rarefied each sample to 5,000 sequences and repeated analyses on 100 random rarefactions. Re-analysis did not quantitatively change results and so we report only the result of the analysis of a single random rarefaction. We included the resulting 275,000 sequences in all subsequent analyses.

## Statistical analyses

We created a database excluding OTUs represented fewer than 3 times to minimize the presence of spurious OTUs caused by PCR and sequencing errors (*Acinas et al., 2005*). We identified the OTUs that were present on all samples to define the "core microbiome" (*Shade & Handelsman, 2012*). Then we tested for significant associations between bacterial taxa and host species, and canopy location using the Linear Discriminant Analysis Effect Size (LEfSe) algorithm (*Segata et al., 2011*). This analysis allows the recognition of significant individual host-microbe associations and evaluates the strength of associations between organisms from different groups (*Segata et al., 2011*).

We performed analyses with the ape (*Paradis et al., 2004*), picante (*Kembel et al., 2010*), and vegan (*Oksanen et al., 2007*) packages in R (*R Development Core Team, 2013*) and ggplot2 (*Wickham, 2009*) for data visualization. We quantified the taxonomic variation in bacterial community structure among samples with the Bray–Curtis dissimilarity. To illustrate patterns of bacterial community structure, we performed a nonmetric multidimensional scaling (NMDS) ordination of Bray–Curtis dissimilarity. We identified relationships between bacterial community structure, host species identity, and sample canopy location by conducting a permutational multivariate analysis of variance (PERMANOVA, *Anderson, 2001*) on the community matrix. We employed a blocking randomization to account for the non-independence of observations among sites. To decompose the total variation in the community matrix explained by host species identity and canopy location, we performed a partial redundancy analysis (RDA; *Legendre & Legendre, 1998*). This technique measures the amount of variation that can be attributed exclusively to each set of explanatory variables. We performed three permutational tests of multivariate homogeneity of group dispersions (Levene's test for variances' homogeneity multivariate equivalent; *Anderson, 2006*; *Anderson, Ellingsen & McArdle, 2006*): one to test if variance in intra-individual canopy bacterial communities was equal between individuals (30 samples from five trees sampled at six canopy locations); a second to compare interspecific variation between species (30 bottom-canopy samples from 30 different trees); and finally a third to test per-species intra- and inter-individual variation (all 55 samples). We estimated phyllosphere bacterial alpha-diversity using the Shannon index calculated from OTU relative abundances for each community. We performed an analysis of variance (ANOVA) and subsequent post-hoc Tukey's tests to compare differences in diversity across species. The authors declare that the experiment comply with the current laws of the country in which the experiment was performed.

## RESULTS

### Sequences, OTUs and taxonomy

High-throughput Illumina sequencing of the bacterial 16S rRNA gene (*Claesson et al., 2010*) identified 5,005 bacterial operational taxonomic units (OTUs, sequences binned at 97% similarity) in the phyllosphere of five temperate tree species, an average of $1,055 \pm 57$ OTUs (mean $\pm$ SE) per tree sampled. Most of these bacterial taxa were relatively common across samples, with only 3.4% of OTUs occurring on a single tree and 0.8% of OTUs

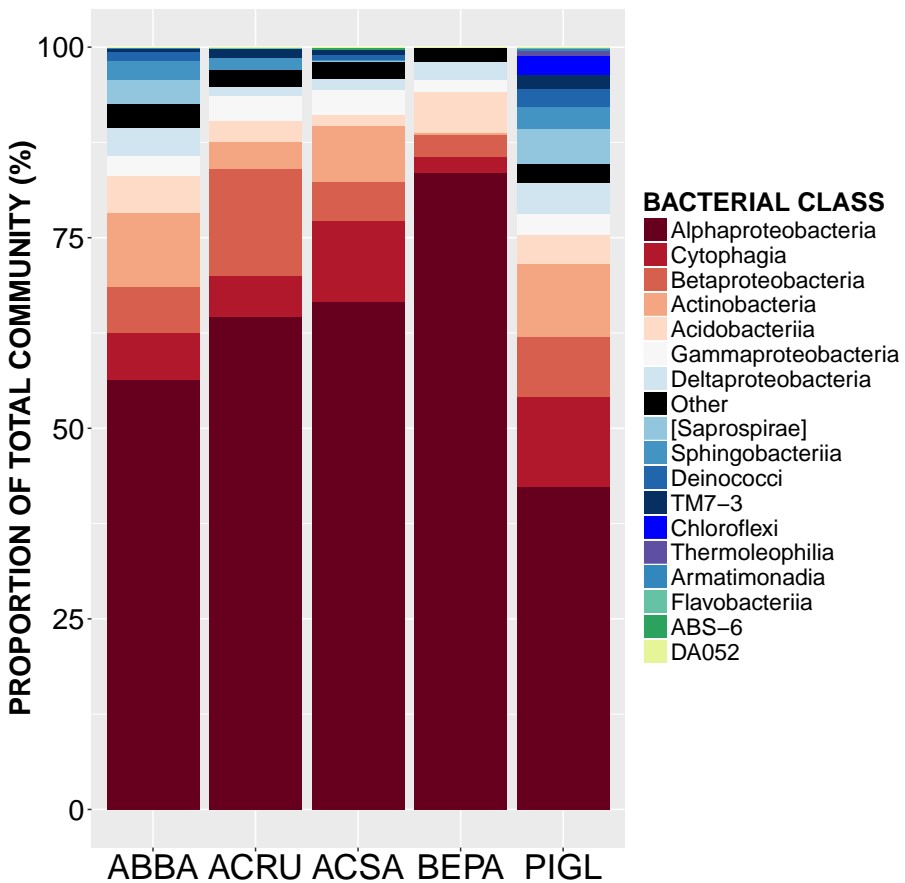

**Figure 1** **Relative abundance of sequences from bacterial taxonomic classes in the phyllosphere microbiome of temperate tree species in Quebec temperate forest.** ABBA, *Abies balsamea*; ACRU, *Acer rubrum*; ACSA, *Acer saccharum*; BEPA, *Betula papyrifera*; PIGL, *Picea glauca*.

occurring on all trees. The OTUs present on all samples represent the "core microbiome": the microbial taxa shared among multiple communities sampled from the same habitat (*Shade & Handelsman, 2012*). In this study, the core  microbiome consisted of 42 OTUs (Table 1) representing 61% of all sequences, of which 72% were *Alphaproteobacteria*, 9% *Cytophagia*, 7.8% *Betaproteobacteria*, 5% *Acidobacteria*, 2% *Gammaproteobacteria* and 2% *Actinobacteria*. The most abundant order was *Rhizobiales* (49%) from which 77% of sequences were assigned to the family *Methylocystaceae*. While there was some variation in the most abundant classes both across the five tree species and among canopy locations (Figs. 1 and 2), the class *Alphaproteobacteria* was always the dominant taxon, with relative abundances ranging from 42% on *P. glauca* to 84% on *B. papyrifera* (Fig. 1).

## Intra-individual vs. inter-individual and interspecific variation

Host species identity and individual identity effects could not be distinguished statistically due to the fact that analyses of intra-individual variation were based on a single individual per species. This host species/individual effect explained 65% of variation in phyllosphere bacterial taxonomic community structure while the impact of canopy location was not

**Table 1  Taxonomy and relative abundance of the 42 OTUs constituting the tree phyllosphere bacterial core microbiome in Quebec temperate forest (present in all 55 samples).**

| Class | Order | Family | Genera | Species | % |
|---|---|---|---|---|---|
| Acidobacteriia | *Acidobacteriales* | *Acidobacteriaceae* | Bryocella | elongata | 0.5 |
| | | | 4 NAs | | 4.8 |
| *Actinobacteria* | Actinomycetales | *Frankiaceae* | NA | | 1.3 |
| | | Microbacteriaceae | Frondihabitans | cladoniiphilus | 0.5 |
| *Cytophagia* | *Cytophagales* | *Cytophagaceae* | Hymenobacter | 2 NAs | 9.0 |
| Sphingobacteriia | Sphingobacteriales | Sphingobacteriaceae | Mucilaginibacter | daejeonensis | 0.5 |
| | | | NA | | 0.2 |
| *Alphaproteobacteria* | *Caulobacterales* | Caulobacteraceae | NA | | 1.5 |
| | *Rhizobiales* | Beijerinckiaceae | Beijerinckia | 2 NAs | 8.9 |
| | | *Methylobacteriaceae* | Methylobacterium | 2 NAs | 2.3 |
| | | *Methylocystaceae* | 7 NAs | | 38.1 |
| | Rhodospirillales | Acetobacteraceae | 6 NAs | | 11.2 |
| | *Rickettsiales* | NA | NA | | 0.10 |
| | | Rickettsiaceae | Rickettsia | NA | 0.6 |
| | Sphingomonadales | Sphingomonadaceae | Sphingomonas | 6 NAs | 7.9 |
| | | | | wittichii | 1.7 |
| | | | | wittichii | 0.1 |
| *Betaproteobacteria* | Burkholderiales | Oxalobacteraceae | 2 NAs | | 7.8 |
| *Deltaproteobacteria* | Bdellovibrionales | Bdellovibrionaceae | Bdellovibrio | NA | 0.2 |
| | Myxococcales | Cystobacterineae | NA | | 0.7 |
| *Gammaproteobacteria* | Enterobacteriales | Enterobacteriaceae | Erwinia | NA | 0.7 |
| | Pseudomonadales | Pseudomonadaceae | Pseudomonas | fragi | 1.3 |

**Table 2  Variation in phyllosphere bacterial community structure explained by various drivers: host species identity, sample location within the tree canopy and individual identity.** PERMANOVA on Bray–Curtis dissimilarities.

| Dataset | Scope | Nb samples | Nb ind./species | Variables $R^2$(%) | | |
|---|---|---|---|---|---|---|
| | | | | Canopy location | Host species identity | Individual identity |
| #1 | Intra-individual | 30 | 1 | 8[a] | 65[b] | |
| #2 | Inter-individual and interspecific | 30 | 6 | *na* | **47** | *na* |
| #3 | Intra- and inter-individual, and interspecific | 60 | 6 | **6** | **47** | **32**[c] |

**Notes.**
[a]The effect of canopy location was not significant after accounting for individual identity.
[b]Host species identity and individual identity are confounded as there were no replicates per species.
[c]Individual identity was nested in host species identity.
*na*, Non applicable.

statistically significant (PERMANOVA on Bray–Curtis dissimilarities; Table 2). We then tested whether canopy position had an effect on community structure after accounting for the variation explained by host species/individual using a partial redundancy analysis (RDA) on bacterial community structure constrained by host species identity. The RDA showed that when differences in bacterial community structure driven by host species identity

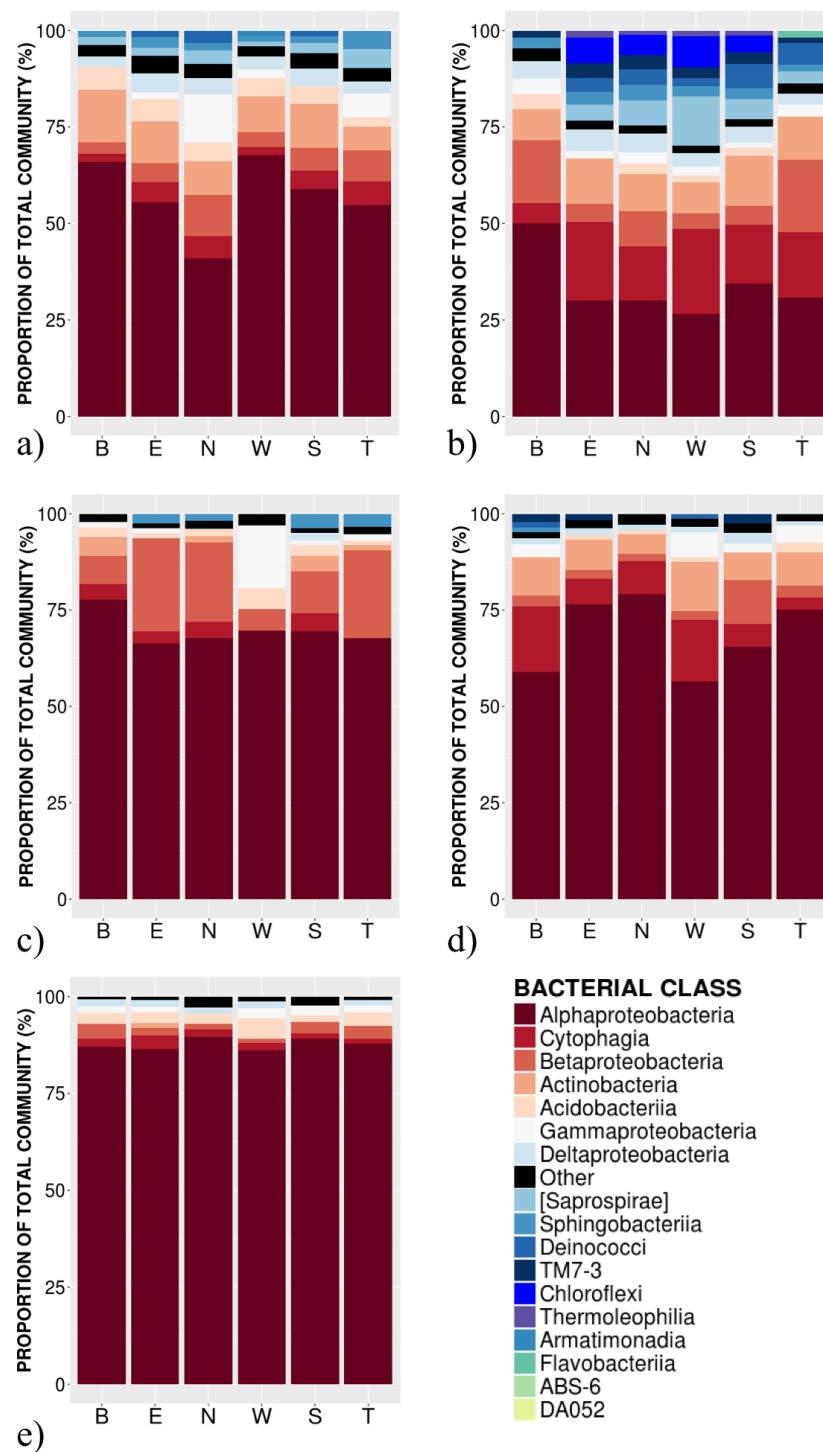

**Figure 2** Relative abundance of bacterial classes in the phyllosphere at 6 canopy locations (B:Bottom, E:East, N:North, W:West, S:South T:Top) for one individual of the five temperate tree species under study. (A) *Abies balsamea*; (B) *Picea glauca*; (C) *Acer rubrum*; (D) *Acer saccharum*; and (E) *Betula papyrifera*.
were accounted for, sample canopy location explained 22% of the remaining variation in community structure. In comparison, in the dataset with 30 different individuals, host species identity explained only 47% of variation in phyllosphere bacterial community structure (PERMANOVA on Bray–Curtis dissimilarities; Table 2). When considering intra-individual and inter-individual samples, host species identity ($R^2 = 47\%$) was the strongest driver of variation in phyllosphere bacterial community structure closely followed by individual identity ($R^2 = 32\%$) and finally by canopy location ($R^2 = 6\%$; PERMANOVA on Bray–Curtis dissimilarities; Table 2). Community composition of samples clustered based both on the individual (Fig. 3A) and species (Fig. 3B) from which they were collected (non-metric multidimensional scaling (NMDS) based on Bray–Curtis distances among samples).

The first permutational multivariate test of variance homogeneity (an analogue of Levene's test of homogeneity of variances) on intra-individual phyllosphere communities indicated a significant difference between *P. glauca* and *B. papyrifera* (Tukey's post hoc test; $P = 0.03$). The second test of the homogeneity of inter-individual variance between host species showed that *P. glauca*'s variance in community structure (mean distance to centroid $= 0.34$) was higher than *A. saccharum* ($0.25$; $P < 0.01$) and *A. rubrum* ($0.26$; $P < 0.05$) while all other comparisons were not significant. Finally, the third test between per species intra-individual and inter-individual variation indicated one significant difference in variation for *B. papyrifera* ($P = 0.005$; Fig. 4).

The alpha-diversity of leaf bacterial community differed significantly across host species identity but not across canopy locations. Post-hoc Tukey honestly significant differences tests confirmed that Shannon alpha-diversity is higher on conifer species ($4.9 \pm$ standard error (SE) of $0.04$ for *A. balsamea* and $5.3 \pm$ SE $0.04$ for *P. glauca*) than on angiosperm species ($3.7 \pm$ SE $0.06$ for *A. rubrum*, $4.1 \pm$ SE $0.05$ for *A. saccharum* and $3.6 \pm$ SE $0.09$ for *B. papyrifera*).

## Bacterial indicator taxa

The LEfSe analysis successfully identified indicator taxonomic groups associated with different host species, but not across different canopy locations (Table 3). The conifers, *A. balsamea* and *P. glauca*, had the highest number of associated bacterial indicator taxa (46 and 188 respectively). The strongest bio-indicators of *A. balsamea* were the *Frankiaceae* family and multiple taxonomic levels of the phylum *Acidobacteria*: *Acidobacteria*, *Acidobacteriales* and *Acidobacteriaceae*. For *P. glauca*, the strongest bioindicators were multiple taxa from the *Bacteroidetes* phylum (*Cytophagia*, *Cytophagales*, *Cytophagaceae*, *Spirosoma* and *Saprospirae*, *Saprospirales*, *Chitinophagaceae*), and from the *Actinobacteria*, *Chloroflexi*, and *Deltaproteobacteria*. In contrast, *B. papyrifera* showed an overrepresentation of 24 bacterial taxa including the phylum *Proteobacteria*, the class *Alphaproteobacteria* and several of its orders (*Rhodospiralles*, *Rickettsiales*, *Caulobacterales*). Finally, the two *Acer* species (*A. rubrum* and *A. saccharum*) were associated with 19 and 32 indicators respectively, including the order *Rhizobiales*: *A. rubrum* being associated with the family *Methylocystaceae* and *A. saccharum* with the order *Methylobacteriaceae*.

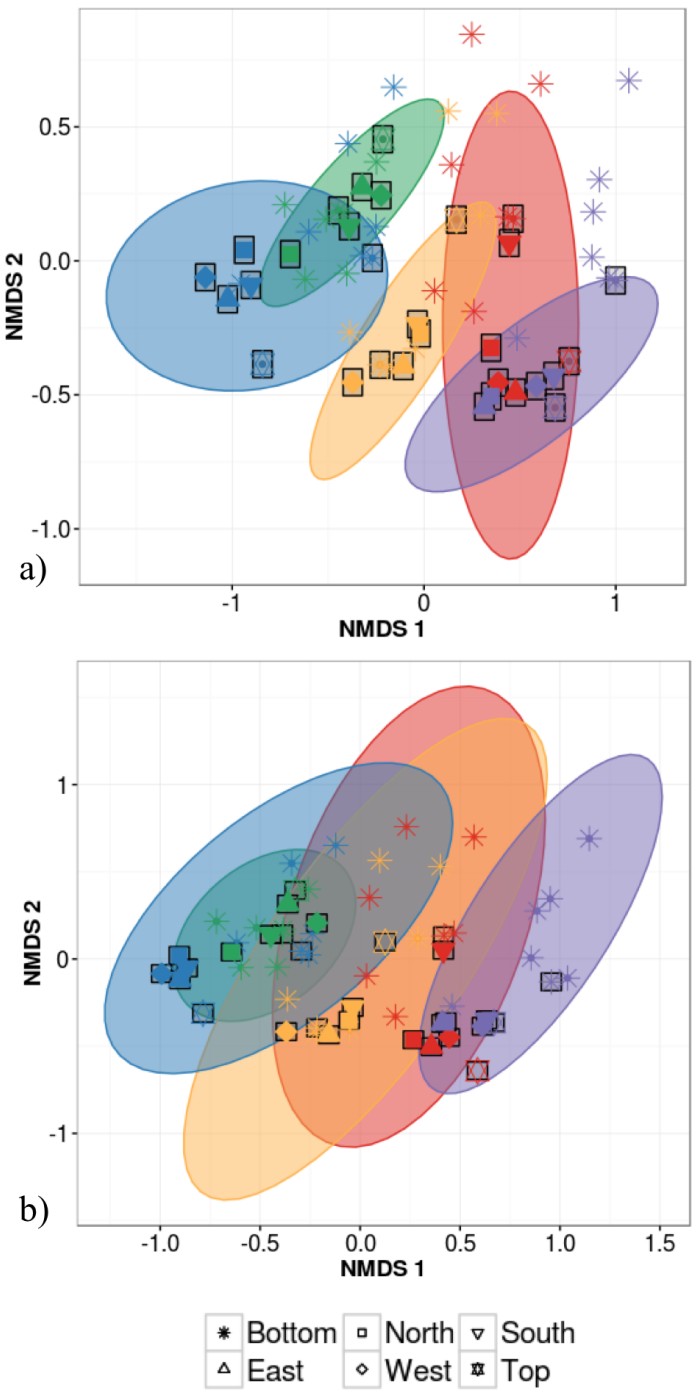

| ✳ Bottom | ▫ North | ▽ South |
| △ East | ◇ West | ✿ Top |

**Figure 3** **Non-metric multidimensional scaling (NMDS) ordination of within-individual variation in bacterial community structure across 55 phyllosphere samples from Quebec temperate forest trees.** Stress amounted to 0.16. Ellipses indicate 1 standard deviation confidence interval around of (A) intra-individual samples and (B) inter-individual samples. Gray boxes indicate the 30 samples that came from individuals sampled at six different canopy locations. The other 25 samples came from 5 more individuals per host species. Symbols indicate sample position in the tree canopy; colours indicate by host species identity (green: *Abies balsamea*; red: *Acer rubrum*; orange: *Acer saccharum*; purple: *Betula papyrifera*; blue: *Picea glauca*).

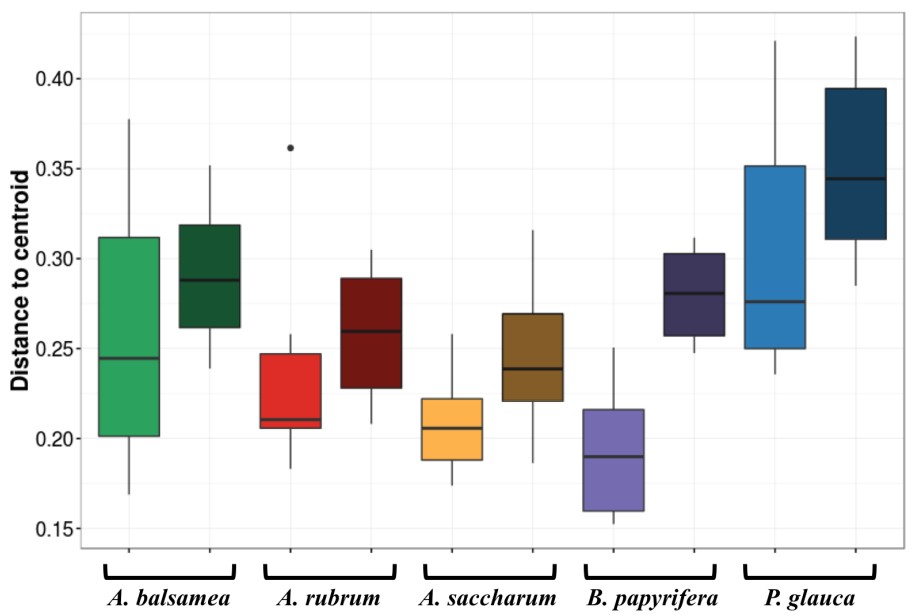

**Figure 4 Permutation test for homogeneity of multivariate dispersions in leaf bacterial communities between per species intra- and inter-individual samples.** Colours indicate host species identity (green for *Abies balsamea*; red for *Acer rubrum*; orange for *Acer saccharum*; purple for *Betula papyrifera*; and blue for *Picea glauca*); shading indicate intra- (pale color) and inter-individual (dark color) variance respectively.

## DISCUSSION

In this study, we demonstrate for multiple host species that there is a significant amount of intra-individual variation in phyllosphere bacterial community structure (Fig. 3A). While the mean distance to centroid is always smaller for intra- than for inter-individual variation (Fig. 4), this distance was only statistically significant for *B. papyrifera*. This result therefore provides partial support for our first hypothesis, stating that magnitude of intra-individual variation would be smaller than inter-individual and interspecific variation. When analyzing all samples, we found host species identity to be a stronger determinant of phyllosphere bacterial community structure than individual identity (Table 2). However, this result could be biased by the fact that we sampled a single individual for multiple canopy location. The importance of host species identity as a driver of phyllosphere community structure agrees with past studies of tropical (*Kim et al., 2012*; *Kembel et al., 2014*; *Lambais, Lucheta & Crowley, 2014*) and temperate trees (*Redford et al., 2010*). Previous studies have quantified intra- and inter-individual variation in phyllosphere bacterial community structure, but these studies mixed leaves from within tree canopies without quantifying intra-individual variation (*Redford et al., 2010*) or explored intra-individual variation for a single host species (*Leff et al., 2015*). Our results show that after taking host species identity into account, there exist detectable differences in microbial community structure within tree canopies, at least in natural forest settings.

In terms of the taxonomic composition of the tree phyllosphere, each tree species can be characterized by a particular combination of most abundant classes across all canopy locations, consistent with other studies of the phyllosphere microbiome

**Table 3  Bacterial taxa identified as bio-indicators of different host species in Quebec temperate forest.** The LEfSe analysis was performed on 30 samples: 6 individuals per species. Only the top five bio-indicators are shown.

| Host species identity | Bacterial taxa | Effect size |
| --- | --- | --- |
| | Actinobacteria.Actinobacteria.Actinomycetales.Frankiaceae | 4.34[***] |
| | Acidobacteria | 4.30[***] |
| *Abies balsamea* | Acidobacteria.Acidobacteriia.Acidobacteriales.Acidobacteriaceae | 4.27[***] |
| | Acidobacteria.Acidobacteriia.Acidobacteriales | 4.27[***] |
| | Acidobacteria.Acidobacteriia | 4.27[***] |
| | Proteobacteria.Alphaproteobacteria.Rhizobiales.Methylocystaceae | 5.13[***] |
| | Proteobacteria.Betaproteobacteria | 4.79[***] |
| *Acer rubrum* | Proteobacteria.Betaproteobacteria.Burkholderiales | 4.79[***] |
| | Proteobacteria.Betaproteobacteria.Burkholderiales.Oxalobacteraceae | 4.77[***] |
| | Proteobacteria.Alphaproteobacteria.Rickettsiales.Rickettsiaceae | 3.81[***] |
| | Proteobacteria.Alphaproteobacteria.Rhizobiales | 5.18[***] |
| | Bacteroidetes.Cytophagia.Cytophagales.Cytophagaceae.Hymenobacter | 4.48[***] |
| *Acer saccharum* | Proteobacteria.Alphaproteobacteria.Rhizobiales.Beijerinckiaceae | 4.47[***] |
| | Proteobacteria.Alphaproteobacteria.Rhizobiales.Beijerinckiaceae.Beijerinckia | 4.47[***] |
| | Actinobacteria.Actinobacteria.Actinomycetales.Microbacteriaceae | 4.33[***] |
| | Proteobacteria.Alphaproteobacteria | 5.39[***] |
| | Proteobacteria | 5.28[***] |
| *Betula papyrifera* | Proteobacteria.Alphaproteobacteria.Rhodospirillales | 5.26[***] |
| | Proteobacteria.Alphaproteobacteria.Rhodospirillales.Acetobacteraceae | 5.25[***] |
| | Proteobacteria.Alphaproteobacteria.Rickettsiales | 4.13[***] |
| | Bacteroidetes | 4.97[***] |
| | Bacteroidetes.Cytophagia.Cytophagales | 4.74[***] |
| *Picea glauca* | Bacteroidetes.Cytophagia | 4.74[***] |
| | Actinobacteria | 4.73[***] |
| | Bacteroidetes.Cytophagia.Cytophagales.Cytophagaceae | 4.73[***] |

**Notes.**
[***] $P < 0.001$.
NS, $P > 0.05$.

(*Redford et al., 2010*; *Kembel et al., 2014*; *Laforest-Lapointe, Messier & Kembel, 2016*). Amongst the potential mechanisms that could explain host species selective power on their phyllosphere bacterial communities, ecological strategies could play a role by impacting leaf abiotic conditions. *B. papyrifera*, a shade intolerant species (*Krajina, Klinka & Worrall, 1982*; *Burns & Honkala, 1990*) exposed to sunlight in the upper part of the forest canopy, exhibited the smallest alpha-diversity with a dominance of *Alphaproteobacteria* (Fig. 2E) and also the smallest amount of intra-individual variation (Fig. 4). In contrast, both conifer host species, growing below a deciduous canopy, exhibited the highest diversity in their community structure. While ultraviolet radiation could be driving the observed differences in leaf alpha-diversity across species, our results provide no evidence of a significant and consistent difference in the alpha-diversity among canopy locations. However, because we sampled only one individual per species, canopy location effects remain to be quantified across multiple individuals of the same species. As shown by the multivariate test of homogeneity of variance, the intra-individual variation in phyllosphere

community structure is not different from the variation observed at the inter-individual level. Future phyllosphere studies characterizing the relative influence of potential key factor such as random colonization via vectors as the atmospheric air flow (*Barberán et al., 2015*) or animals (*Scheffers et al., 2013*), competition between bacterial populations (*Vorholt, 2012*); or intra-individual variation in leaf functional traits (*Hunter et al., 2010*; *Reisberg et al., 2012*) are needed to understand the dynamics driving intra-individual variability in bacterial community structure.

In conclusion, our results demonstrate that there exists considerable intra-individual variation in phyllosphere community structure, and that the magnitude of this variation is smaller but not statistically different from the magnitude of inter-individual variation. When designing a study of tree phyllosphere bacterial communities, if quantifying interspecific variation is the goal then samples from a consistent location within the tree canopy for individual trees are sufficient to quantify the majority of the variation in community structure. However, future studies and especially studies focusing on a single host species should acknowledge that there can be significant intra-individual variation in phyllosphere community structure, and sampling plans should explicitly select leaves at different positions within the canopy to describe spatial structure of the overall community composition for individual trees.

## ACKNOWLEDGEMENTS

We thank Travis Dawson, Sophie Carpentier and Gabriel Jacques for support in the field and laboratory.

### Funding

Financial support was provided by the Natural Sciences and Engineering Research Council of Canada (NSERC), the Fonds de Recherche du Québec - Nature et Technologies (FRQNT), and by the Canada Research Chairs Program. The funders had no role in study design, data collection and analysis, decision to publish, or preparation of the manuscript.

### Grant Disclosures

The following grant information was disclosed by the authors:
Natural Sciences and Engineering Research Council of Canada (NSERC).
Fonds de Recherche du Québec - Nature et Technologies (FRQNT).
Canada Research Chairs Program.

### Competing Interests

The authors declare there are no competing interests.

## Author Contributions

- Isabelle Laforest-Lapointe conceived and designed the experiments, performed the experiments, analyzed the data, wrote the paper, prepared figures and/or tables, reviewed drafts of the paper.
- Christian Messier conceived and designed the experiments, reviewed drafts of the paper.
- Steven W. Kembel conceived and designed the experiments, analyzed the data, contributed reagents/materials/analysis tools, wrote the paper, reviewed drafts of the paper.

## Data Availability

laforest-lapointe, isabelle (2016): Code for INTRA_MS_peerJ. figshare.
https://dx.doi.org/10.6084/m9.figshare.3370021.v1

laforest-lapointe, isabelle (2016): OTUs_biom_in_csv_intraindividual. figshare.
https://dx.doi.org/10.6084/m9.figshare.3178837.v1

laforest-lapointe, isabelle (2016): Barcodes. figshare.
https://dx.doi.org/10.6084/m9.figshare.2062521.v2

laforest-lapointe, isabelle (2016): Metadata_intraindividual. figshare.
https://dx.doi.org/10.6084/m9.figshare.3178795.v1

laforest-lapointe, isabelle (2016): Sequences. figshare.
https://dx.doi.org/10.6084/m9.figshare.2062512.v1.

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
