# Peer review of "Tree phyllosphere bacterial communities: exploring the magnitude of intra- and inter-individual variation among host species"

_PeerJ, doi:10.7717/peerj.2367_

## Round 0.1 · original submission · Major Revisions

The reviewers have provided an unusually detailed set of recommendations for your paper. While all agree as to the significance and timeliness of this research, they have also indicated some ways in which a thorough revision might improve the manuscript.

I agree that due to lack of replication (with the exception of the bottom canopy) interpretation of among-species variances, and some canopy location comparisons are unfounded here. Accounting for this will require a thorough re-analysis, and in my opinion, revised conclusions.

I look forward to your novel insights in the next submission.

·

Basic reporting

Introduction
L104-106 the data in this reference does not really support this statement. Furthermore, while there is evidence (not cited here) for phototrophic organisms in the phyllosphere, they do not require UV radiation. Please re-write this sentence.

Methods
L170 this sentence should be moved up to L166 as I assume the data was rarefied and cleaned prior to OTU clustering.
L184 should clarify that ggplot2 was used for data visualization, not analysis.

Experimental design

No Comments

Validity of the findings

No Comments

Additional comments

This manuscript describes intra and inter individual variation in the bacterial commnity structure of the leaf surfaces of five trees from a temperate forest. The article is presented in a clear and succinct fashion, the data is all available to the reader and it is presented with in the broad context of the field. I have only some minor comments that I listed under "Basic Reporting".

Reviewer 2 ·

Basic reporting

Acceptable

Experimental design

Acceptable (but see General Comments for limitations)

Validity of the findings

Acceptable

Additional comments

The study attempts to address the degree of different levels of variation (intraindividual, intraspecific, and interspecific) in the phyllosphere community of trees. While that’s an interesting question, I’m not convinced that the study is particularly well designed to address it. The intraindividual aspect is generally OK – six different leave collections were sampled from each tree, from different positions within the canopy. However I’m not convinced that intraspecific and interspecific differences are really assessed well at all. Just a single individual tree from each species was examined in detail, with two extra trees per species sampled by a single leaf. Is sampling three individuals enough to determine intraspecific variation or even interspecific variation? Given the accessibility of NGS methods, I would say not. It’s a start, but the limitations of minimal replication need to be addressed much more strongly. Sampling is also limited to one location on one day and phyllosphere communities are known to show temporal and biogeographic variation within a species. Thus, the sampling design is inherently selecting for reduced intraspecific variation by eliminating those factors – it’s not surprising to find that host species is the most important when variables that could influence intraspecific variation are minimized, and sampling for that variation is minimal. That there is variation between locations on a tree has also been previously shown (I agree with the authors that it’s often ignored), and while this study confirms that, it can’t really be linked canopy location because there is no replicate sampling within a particular canopy location on an individual tree. If a study was to really assess the influence of canopy position on the phyllosphere, one would sample replicate leaves from multiple locations within the canopy and see if there is more variation between canopy positions than within them. Similarly, more individuals of each species really need to be sampled to get a better feel for the levels of individual and species level variation. These limitations don’t invalidate this study, but the authors need to more clearly acknowledge them and recognize that some of their conclusions and arguments are presented more strongly than is merited by the study design. I’d even suggest that the title exaggerates the findings more than necessary, given the study limitations.

Abstract
26-29 – Is this correct? It’s more relevant to the Introduction but the authors state that other studies do this without really citing any. All of my phyllosphere work has used multiple leaves per plant, so I’m not convinced that this problem is as pronounced as stressed here.
33-38 – It needs to be clearly stated that just one individual tree per species was sampled.
35 – see comment in the text on “randomly”. Was it random?
41-42 – Authors need to be careful about attributing variation to tree species when the limited sample size means that it’s individual to individual variation (albeit those individuals are different species)
Introduction
64-66 – It would be helpful to cite some studies that “assume” this or do this. As it is, it reads more like the authors asserting this.
72-75 – Again, cite some of these offending studies. The authors cite more papers that are exceptions
83 – The importance of interspecific variation is quite well known as we know that taxonomy/genetics (i.e. species differences) are important in the phyllosphere.
87-100 – This paragraph seems out of place and of limited use here (it reads more like it was pasted in from another manuscript). The phyllosphere and its importance were introduced earlier and the focus should now be on variation. It could probably be deleted.
109-114 – Atmosphere conditions within the canopy also likely have an effect. Internal leaves are likely exposed to higher moisture levels in the surrounding air and potentially different gas levels in densely canopied species.
116-117 – As mentioned in the general comments, is focusing at one site really a good measure of intraspecific variation (esp. given the limited number of individuals)
Materials and Methods
130 – Table 1 doesn’t seem relevant here
135-142 – Very little information is given about the trees sampled. For a study examining variation like this, I would expect information on extent of canopy cover, light irradiance, slope, aspect, tree age (or at least DBH), height to be presented. These are all factors that could potentially influence variation between trees (intra- or interspecific). It would also be useful to know how dominant these trees are at this site (i.e. what % of the forest does each species account for). How were trees “randomly chosen”? Ecologists typically go through specific randomization processes during sampling, but I don’t see that here. If it was just “pick that tree” then that’s not random (haphazard maybe?). This applies to line 138 and 141.The broader issue is that with only one individual of each species being sampled in detail, then assessing interspecific variation is really assessing individual variation.
139-144 – How many leaves were sampled? Surface area? Mass isn’t really meaningful here. Was it standardized per “sample” (50-100 g is a large range and more diverse communities could be assumed to come from a greater mass). Why not just sample a single leaf if the idea is to look at variation within a single individual?
144-145 – More details on the phyllosphere removal step would be useful here. “Agitating” is very vague – how was that done, speed, standardized time etc.
150-160 – A two-step PCR severely limits the study here as it compounds any amplification biases. Stating that it was done as it “reduced the number of primers” (line 152) is a ridiculous justification. Using a single amplification step this entire study could have been amplified with just 13 primers using a 5 x 8 paired end system. In any case, the second step of the two-step PCR is never presented.
177 – Should we even expect a “core microbiome” for multiple tree species? Far more useful would be identifying a core microbiome per species….but that would require more individuals of each species to be sampled.
Results
209 – abundant? No measures of abundance were made.
212 – Again, the “core microbiome”. Should we regard different tree species as the same habitat? Given the species level variation I would argue against that, which makes the core microbiome idea less useful here.
217 – Here and elsewhere (e.g. 222, 224, 234), care needs to be taken that referring to differences across the five tree species is largely limited to comparisons between five individuals
247-248 – “a fact also confirmed by”. This is poor writing. The finding (not fact) is confirmed (or better yet, supported), but in any case, this is arguing that one metric is supported by another, when actually they are related (i.e. that the presence of multiple bacterial taxa on certain samples but not others will likely raise the diversity of that community). Also, Fig. 2 (cited here) doesn’t show this. Regardless, why not report OTU diversity here? That would be a more direct assessment of alpha diversity.
255 – “differentially abundant”. Huh?

Discussion
This section really needs more expansion on the limitations of this study (see above) and some potential problems in data interpretation between tree species or individual etc. I’d suggest a thorough reworking as the current discussion is quite short and could be expanded. Some specifics
286 – use past tense (exhibited not exhibits)
291-292 – Without sampling canopy position in multiple individuals per species (and potentially multiple canopy position replicates) these statements are very limited
296 – This is a vague statement (not to sample top canopy leaves) attributing the reason to an older study. If this is a recommendation, be more explicit in why not to do this.
296-301 – Yes, I’d agree with all of these. But don’t we already know that we need more studies like this?
303-306 – Again, acknowledge the limitations here. This conclusion is fairly strong but based on limited sampling.

Figure 1 – I’m not sure that panel b adds much here. It might also be better to abbreviate trees like “A. ba” or “A. ru” instead of all caps as that relates better to a Genus species name (applies to all figures). It’s also confusing to look at the color key and read down it whereas the charts read up (i.e. Acidobacteria are at the top of the key, but the bottom of the charts) – flip the key. Better yet, this would look better if organized by relative proportion of the community rather than alphabetically (i.e. organize the “classes” by importance)
Figure 2 – Clarify that this is from a single individual of each tree. Organize data as suggested in comments on Figure 1.
Figure 3 – Report stress as it identifies how well the NMDS ordination works. As with other figures, clarify that the majority of samples came from just one individual per species. There’s really no need to present the abbreviations for tree species here as the key in the figure could list the full species names. Legend should more clearly identify the samples which came from other trees (I’m assuming the boxed * points represent the two additional trees sampled from the bottom canopy, but it isn’t clear). Regarding those, there is quite a bit of variation – as much as within a canopy for some species.
Figure 4 – As with Fig 3, the key could include full tree names. Clarify in legend that it is a single individual per species.
Table 1 – As with the figures, clarify the number of samples in the table header and tree species. I’m still not convinced that the “core microbiome” is of much use here.
Table 2 – Delete. All of this information is presented in the text.
Table 3 – Clarify that it’s primarily based on one individual per species in the header. There is no need for the *** column. Just write in the table header that all were significant at P<0.001.

Reviewer 3 ·

Basic reporting

Phyllosphere microbial communities vary within the canopy of the same tree, among individuals of the same host species and among host species. The goal of the study was to assess these variations, compare them, and derive practical recommendations for leaf sampling. These recommendations will be very useful for all future studies on the phyllosphere.

Experimental design

However, I am concerned with the sampling design which was chosen to reach this aim. It seems that within-canopy variations were only assessed for one tree per host species. Because of this lack of replication, the effect of host species and tree individual are confounded in some analyses and figures. The authors should discuss more thoroughly the limitations of the sampling design. The literature should also be updated and better chosen.

Validity of the findings

See above

Additional comments

***Abstract

The abstract is very clearly written but a few sentences could be improved:

Line 35, "For randomly selected trees of each species": how many trees per species were sampled?

Line 37, "We also collected bottom-canopy leaves from two additional trees from each species": does it mean that three trees were sampled?

Line 40, "Based on analysis of bacterial community structure measured via sequencing of the bacterial 16S gene": which sequencing platform was used?

Line 46, "Our results suggest that for interspecific studies of phyllosphere bacteria, individual
samples from consistent positions within the tree canopy can be used": why is the conclusion different for intraspecific studies?

Line 50, "However, for intraspecific studies, it may be necessary to account for variation in phyllosphere microbiome structure within individual trees and host species": why is it necessary to take into account host species in intraspecific studies?

***Introduction

The introduction is also very clearly written but the litterature is not always well chosen:

Line 56: Microbiome is not the right term. Microbiota would be more adequate (see Schlaeppi & Bulgarelli 2015 The plant microbiome at work MPMI)

Line 60: Recent articles on the protective role of phyllosphere microbial communities should be cited.

Line 62: Phyllosphere microbial communities also influence water cycling. Some references should be included.

Lines 64-65: Studies in which a single sample of leaf is considered as representative of the community of the entire tree or the entire host species should be cited here.

Line 70: The study by Horton et al. (2014) published in Nature could be cited here.

Lines 87-90: 2009 is not really the past. Phyllosphere research started in the 50s. The references are not well chosen, many of them focus on phyllosphere fungi.

Line 92: No they are not involved in methane degradation. See Knief et al. 2012.

Lines 95-97: Is it really because of the discovery of the ecological importance of phyllosphere microbes? Isn't it because of the emergence of environmental genomics?

Line 101-114: This is the most important section of the introduction. More references are needed. Within-canopy variations in microclimate should be better described.

Line 117: phyllosphere bacterial communities

Line 123: Why is the immigration rate expected to be higher for light leaves?

***Materials and methods

Line 130: how many trees per species were sampled?

Line 171: how many rarefactions were performed? were the results robust to rarefaction?

***Results

Figure 3: What are the stars in grey squares? Do they correspond to the bottom-canopy leaves from two additional trees from each species? How were they included in the analyses? Were they included in the cluster analysis?

Table 2: The species effect can only be tested for bottom-canopy leaves (because there are replicates per species). I don't think that the effect of species by canopy location can be tested with the dataset.

Figure 4: The legend indicates that the figure is a permutation test. This is not clear because the figure actually represents a multivariate analysis very similar to that of Figure 3 (PcoA instead of NMDS). Moreover the figure is mileading because the host species effect could be a tree effect (there is only one tree per species)

***Discussion

The limitations of the sampling design should be discussed.

Line 270, "intra-individual variation is not significant when compared to the magnitude of interspecific variation" : This is not what I see on Figure 3. Intra-individual variation is huge. The dissimilarity between two samples of the same tree can be larger than that of two samples taken from different tree species.

Lines 28é-289: This section is very speculative.

Line 296, " leaf samples should not include top canopy leaves". It depends on the purpose of the study.

Line 298: Copeland et al. (2015) studied the seasonal dynamics of phyllopshere bacterial communities, not the influence of wind. Other articles should be cited here.

---

## Round 0.2 · accepted · Accept

Thank you for your thoughtful improvements to the manuscript. I appreciate your attention the long list of editor criticism. I look forward to seeing it in print.

Reviewer 2 ·

Basic reporting

No Comments

Experimental design

No Comments

Validity of the findings

No comments

Additional comments

I reviewed an earlier draft of this manuscript and was happy to see this improved version. Thank you for addressing the majority of what was a quite extensive list of suggestions.